# The effect of real-time EF automatic tool on cardiac ultrasound performance among medical students

Noam Aronovitz[1]*, Itai Hazan[1,2], Roni Jedwab[1,2], Itamar Ben Shitrit[1,2,3], Anna Quinn[4], Oren Wacht[5], Lior Fuchs[2,3,6]

1 Joyce and Irving Goldman Medical School, Faculty of Health Sciences, Ben-Gurion University of the Negev, Beer-Sheva, Israel, 2 Clinical Research Center, Soroka University Medical Center and Faculty of Health Sciences, Ben-Gurion University of the Negev, Beer-Sheva, Israel, 3 Department of Epidemiology, Biostatistics and Community Health, Faculty of Health Sciences, Ben-Gurion University, Beer-Sheva, Israel, 4 Medical School for International Health in Beer-Sheva, Beer-Sheva, Israel, 5 Department of Emergency Medicine, Ben Gurion University of the Negev in Beer- Sheva, Israel, 6 Medical Intensive Care Unit, Soroka University Medical Center, Beer-Sheva, Israel

* noamaro@post.bgu.ac.il

**Data Availability Statement:** The minimal data set file we uploaded contains all the data we based our analysis on. The files including the actual POCUS clips cannot be shared publicly as initial ethical

## Abstract

### Purpose

Point-of-care ultrasound (POCUS) is a sensitive, safe, and efficient tool used in many clinical settings and is an essential part of medical education in the United States. Numerous studies present improved diagnostic performances and positive clinical outcomes among POCUS users. However, others stress the degree to which the modality is user-dependent, rendering high-quality POCUS training necessary in medical education. In this study, the authors aimed to investigate the potential of an artificial intelligence (AI) based quality indicator tool as a teaching device for cardiac POCUS performance.

### Methods

The authors integrated the quality indicator tool into the pre-clinical cardiac ultrasound course for 4th-year medical students and analyzed their performances. The analysis included 60 students who were assigned to one of two groups as follows: the intervention group using the AI-based quality indicator tool and the control group. Quality indicator users utilized the tool during both the course and the final test. At the end of the course, the authors tested the standard echocardiographic views, and an experienced clinician blindly graded the recorded clips. Results were analyzed and compared between the groups.

### Results

The results showed an advantage in quality indictor users' median overall scores (P = 0.002) with a relative risk of 2.3 (95% CI: 1.10, 4.93, P = 0.03) for obtaining correct cardiac views. In addition, quality indicator users also had a statistically significant advantage in the overall image quality in various cardiac views.

permission is now outdated, and also because permission was not granted a-priori. Attached, are the credentials and contact information for the chairman of the ethics committee: Abed N. Azab, Ph.D. Associate Professor of Clinical Pharmacology Head of the Ethics Review Board, Faculty of Health Sciences Recanati School for Community Health Professions Faculty of Health Sciences Ben-Gurion University of the Negev P.O.B 653, Beer-Sheva 84105, Israel Phone: (972)-86479880; Fax: (972)-86477683 Email: azab@bgu.ac.il, ethics@medic.bgu.ac.il.

**Funding:** GE Healthcare© provided the POCUS devices used in this study. Lior Fuchs declares that he works as a consultant for GE Healthcare©. However, it's important to note that GE Healthcare© provided support solely in the form of lending the POCUS systems for the research. They did not play any additional roles in the study design, data collection and analysis, decision to publish, or manuscript preparation. The specific roles of Lior Fuchs are detailed in the 'author contributions' section. It's worth highlighting that this research was conducted independently and not in his capacity as a consultant for GE Healthcare©. Additionally, Lior Fuchs did not receive any financial support or salary from GE Healthcare© for the work he contributed to this research.

**Competing interests:** I have reviewed the journal's policy, and the authors of this manuscript have the following competing interests: GE Healthcare© provided the POCUS devices used in this study. Lior Fuchs declares that he is a consultant for GE Healthcare. However, it's important to note that the company had no access to the idea, to the study's primary objective, nor to its design, data analysis, or writing. This affiliation does not affect our adherence to PLOS ONE policies regarding data and material sharing. The remaining authors declare that they have no competing interests.

## Conclusions

The AI-based quality indicator improved cardiac ultrasound performances among medical students who were trained with it compared to the control group, even in cardiac views in which the indicator was inactive. Performance scores, as well as image quality, were better in the AI-based group. Such tools can potentially enhance ultrasound training, warranting the expansion of the application to more views and prompting further studies on long-term learning effects.

## Introduction

Ultrasound imaging is a sensitive, safe, low-cost, and non-invasive tool. Ultrasound devices are more portable due to technological advances, becoming ubiquitous in different clinical settings outside the services of traditionally trained medical imaging specialists. Point-of-care ultrasound (POCUS) is the use of portable ultrasound devices at the bedside by non-radiologists [1].

Recent studies established the importance of integrating POCUS in bedside physical exams [2–4]. POCUS utilization includes cardiac function evaluation, shock diagnosis and management, image-guided procedures, and many other applications. It enhances internal medicine residents' skills, as reflected in research covering diagnostic assessment of left ventricle (LV) function, valve diseases, and LV hypertrophy [5]. A randomized controlled trial showed that early POCUS exams in patients with chest pain and dyspnea reduced time for initiation of appropriate treatments [6].

The improved diagnostic performance and the positive clinical outcomes attributed to POCUS use, highlight the importance of its integration into medical education [7]. Most American medical school curricula now integrate ultrasound training [3, 8]. However, its highly operator-dependent modality requiring experience creates potential difficulty when implementing ultrasound training, especially for POCUS.

A study reviewing the POCUS-guided diagnosis of aortic aneurysms by emergency department physicians showed markedly varied results correlating with user experience [9]. An Australian study presented a distinct correlation between user experience and interobserver agreement with expert echocardiographers in transthoracic hemodynamic POCUS evaluation [10]. Both examples show how essential POCUS integration in medical training is for novice users to gain experience.

Despite the advantages mentioned, POCUS is not yet sufficiently utilized in many clinical settings. One American study conducted in 2020 surveyed POCUS use in all Veterans Affairs medical centers. It showed that the number of physicians using POCUS has not changed significantly between 2015 and 2020 despite the availability of equipment [11]. The low utilization rates among experienced physicians and the limited experience of novice clinicians underscore the importance of integrating high-quality POCUS training early in medical education as well as implementing a feedback system independent of the user's personal experience. To address this gap, it is crucial to focus on enhancing technological solutions, supporting inexperienced users in their clinical practices, and developing efficient training methods. In our study, we aimed to evaluate the effectiveness of one such technological solution, the artificial intelligence (AI) quality indicator tool, by assessing the cardiac ultrasound performance of inexperienced POCUS users in both intervention and control groups.

Some POCUS device manufacturers have already added automatic AI-based tools to enhance the performance and imaging abilities of novice and experienced users alike. These companies include but are not limited to Phillips, GE Healthcare, DiA, Pulsnmore, Kosmos, Ultrasight, and Caption Health. The Real-Time quality indicator is part of the Real-Time Ejection Fraction (EF) tool (by GE Healthcare, Venue POCUS family system) designed to execute automated calculations of EF values in the apical 4-chamber position. The tool provides live quality feedback of the apical 4-chamber image through a superimposed colored left ventricular contour line (red—poor, yellow—moderate, or green–good). The quality is based on AI analysis of image quality (Fig 1).

In our study, we aimed to address the shortage of experienced POCUS operators among medical school graduates. Considering previous research indicating that students often struggle with apical cardiac views [11], we hypothesized that an AI-based quality indicator tool, specifically designed for apical 4- and 5-chamber views, could serve as an effective teaching aid, to improve the cardiac ultrasound skills of novice operators. Our study was designed to compare the success rates and quality of apical views, as well as other cardiac views, between students using the AI-based tool and those using the standard POCUS device, in order to test the added value of integrating such tools in medical education and clinical work.

## Methods

This is a prospective randomized controlled study in medical education, where the reader of the ultrasound test results was blinded to the study groups.

### Study population

The study was conducted at Ben-Gurion University during the pre-clinical cardiac ultrasound course, specifically involving 4th-year medical students in a 6-year program. Recruitment for this study took place on January 22nd, 2022. The possibility to participate was offered to all students who were enrolled in the POCUS course (a mandatory course in the curriculum). Exclusion criteria included previous POCUS experience or training, failure to sign informed consent and failure to observe group allocation during the training. Further exclusion criteria were applied during data processing after the 6-minute exam was conducted, as specified in the results section. Participating students filled out a personal questionnaire (see full questionnaire, S1 Appendix) and signed an informed consent form permitting the use of the data collected in the study for research purposes only. The questionnaire included personal and demographic details, extracurricular POCUS training hours, and thoracic anatomy knowledge reflected by the grade received in an academic course. The study was approved by the ethics committee of Ben-Gurion University Faculty of Health Sciences study. ID– 36–2021, November 28, 2021.

Students were divided into 4- to 6-member training groups. In the first training session, we randomly assigned these groups to one of two POCUS training methods. The first group utilized the AI-based quality indicator tool described below (the Auto-EF tool by Venue GE Healthcare). This student group was defined as the intervention group that underwent training and testing using the quality indicator. The second group, defined as the control group, was trained with standard POCUS systems without using the quality indicator tool. Group profiles were analyzed based on demographics and other relevant variables to ensure random assignment and equal representation [Table 1].

### Point of care cardiac ultrasound training course

All participants completed the same 8-hour frontal POCUS course, comprised of two 4-hour sessions focused on obtaining basic transthoracic cardiac views. The first covered basic

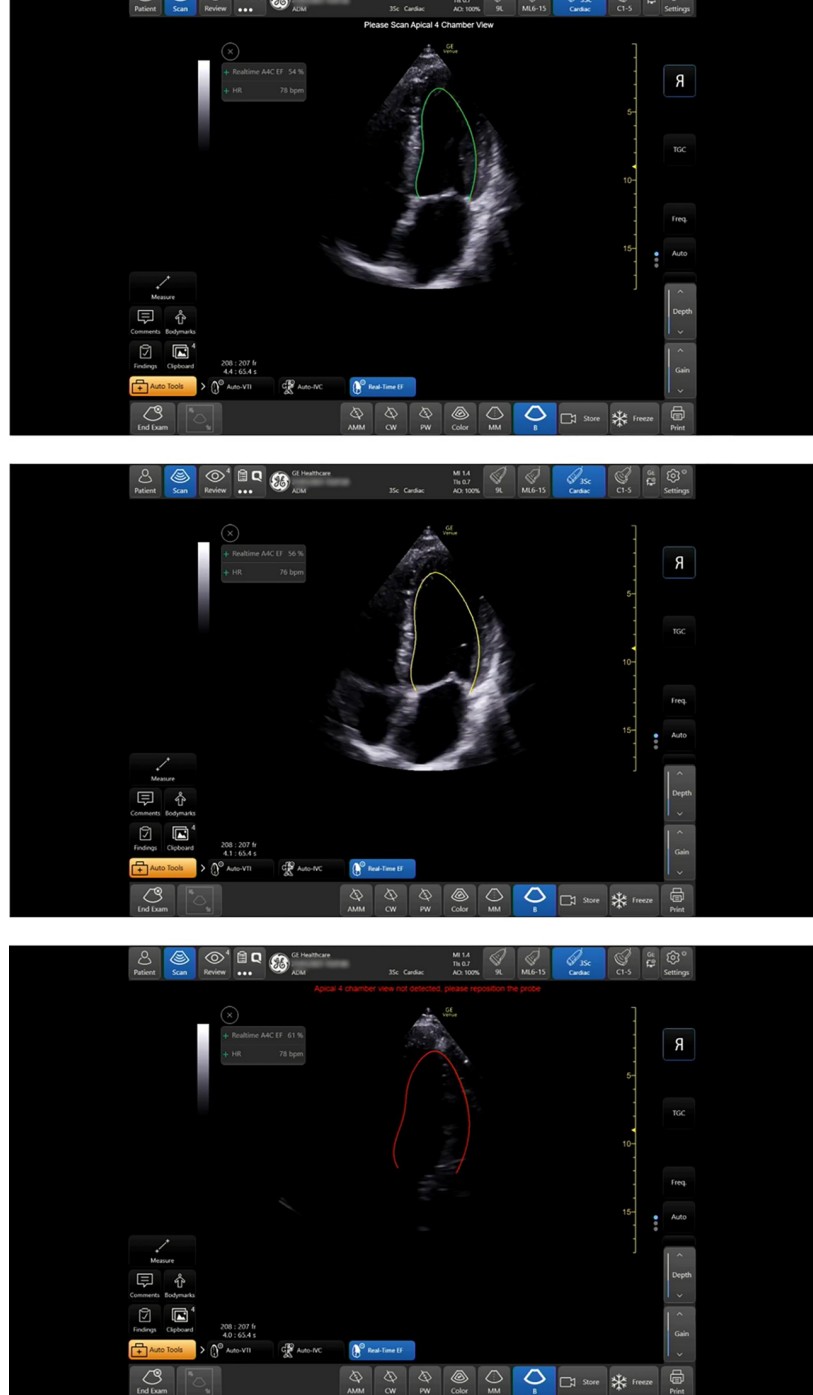

**Fig 1. The quality indicator tool.** Quality indicator contour lines in green, yellow and red, correspond with good, medium and bad quality apical 4 chamber positioning. Republished from [12] under a CC BY license, with permission from GE HealthCare, original copyright 2021.

principles of cardiac ultrasound imaging, sonographic heart anatomy, and common pathologies. The second included hands-on training focused on acquiring various cardiac views. Students were allocated randomly to one of the study groups: the AI-based quality

**Table 1. Comparison of success rates (apical 4/5-chamber excluded) and demographics between control and quality indicator groups, a total of 60 participants, Ben Gurion University, Israel, 2022.**

| Characteristic | | Control, N = 39 | Quality indicator N = 21 | Cohen's D | p-value | Overall, N = 60 |
|---|---|---|---|---|---|---|
| Male sex, n (%) | | 18 (46%) | 12 (57%) | 0.214 | 0.4 | 30 (50%) |
| Age, years, Mean (SD), (N) | | 27.3 (3.8), (38) | 26.9 (2.1), (21) | 0.122 | 0.6 | 27.1 ± 3.3 (59) |
| Ethnicity, n (%) | | | | 0.012 | | |
| Jewish | | 34 (87%) | 18 (86%) | | >0.9 | 52 (87%) |
| Bedouin | | 1 (2.6%) | 1 (4.8%) | | | 2 (3.3%) |
| Non-Bedouin Arab | | 4 (10%) | 2 (9.5%) | | | 6 (10%) |
| Thoracic anatomy grade (0–100%), Mean (SD), (N) | | 89.8 (5.6), (39) | 89.6 (4.2), (20) | 0.047 | 0.6 | 89.8 ± 5.2 (59) |
| Hours of extracurricular training | | | | 0.092 | 0.6 | |
| Mean (SD), (N) | | 1.45 (1.22), (38) | 1.56 (1.04), (21) | | | 1.49 ± 1.15 (59) |
| Median (IQR) | | 1.50 (0.13, 2.00) | 2.00 (1.00, 2.00) | | | 1.50 (0.84, 2.00) |
| Range | | 0.00, 5.00 | 0.00, 4.00 | | | 0.00, 5.00 |
| **Parasternal long view** | Total score, Mean (SD), (N) | 2.77 (1.33), (39) | 3.48 (1.25), (21) | **0.536** | **0.01** | 3.02 ± 1.33 (60) |
| | Correct alignment, n (%) | 31 (79%) | 19 (90%) | 0.289 | 0.5 | 50 (83%) |
| | Total endocardial demarcation, n (%) | 23 (59%) | 19 (90%) | **0.706** | **0.01** | 42 (70%) |
| | Mitral valve visualization, n (%) | 31 (79%) | 18 (86%) | 0.157 | 0.7 | 49 (82%) |
| | Aortic valve visualization, n (%) | 23 (59%) | 17 (81%) | 0.464 | 0.08 | 40 (67%) |
| **Parasternal short, base view** | Total score, Mean (SD), (N) | 2.54 (1.27), (39) | 3.05 (1.07), (21) | 0.416 | 0.13 | 2.72 ± 1.22 (60) |
| | Aorta visualization, n (%) | 32 (82%) | 17 (81%) | 0.028 | >0.9 | 49 (82%) |
| | Tricuspid valve visualization, n (%) | 26 (67%) | 17 (81%) | 0.311 | 0.2 | 43 (72%) |
| | Pulmonic valve visualization, n (%) | 21 (54%) | 12 (57%) | 0.064 | 0.8 | 33 (55%) |
| | Interatrial septum visualization, n (%) | 20 (51%) | 18 (86%) | 0.738 | **0.008** | 38 (63%) |
| **Parasternal short, mitral valve view** | Total score, Mean (SD), (N) | 1.62 (0.63), (39) | 1.48 (0.68), (21) | 0.212 | 0.4 | 1.57 ± 0.65 (60) |
| | Complete LV visualization, n (%) | 29 (74%) | 17 (81%) | 0.152 | 0.8 | 46 (77%) |
| | Mitral valve visualization, n (%) | 34 (87%) | 14 (67%) | 0.513 | 0.09 | 48 (80%) |
| **Parasternal short, mid-papillary view** | Total score, Mean (SD), (N) | 1.33 (0.84), (39) | 1.33 (0.91), (21) | 0 | >0.9 | 1.33 ± 0.86 (60) |
| | Complete LV visualization, n (%) | 24 (62%) | 13 (62%) | 0.007 | >0.9 | 37 (62%) |
| | Papillary muscle visualization, n (%) | 28 (72%) | 15 (71%) | 0.008 | >0.9 | 43 (72%) |
| **Apical 2-chamber view** | Total score, Mean (SD), (N) | 1.54 (1.14), (39) | 1.67 (1.32), (21) | 0.105 | 0.6 | 1.58 ± 1.20 (60) |
| | Open LV (apical 2-chamber), n (%) | 19 (49%) | 10 (48%) | 0.021 | >0.9 | 29 (48%) |
| | Mitral valve anatomy (apical 2-chamber), n (%) | 26 (67%) | 14 (67%) | 0 | >0.9 | 40 (67%) |
| | Open LA (apical 2-chamber), n (%) | 15 (38%) | 11 (52%) | 0.275 | 0.3 | 26 (43%) |
| **Subcostal view** | Total score, Mean (SD), (N) | 2.41 (0.88), (39) | 2.71 (0.46), (21) | 0.394 | 0.3 | 2.52 ± 0.77 (60) |
| | Open RV, n (%) | 35 (90%) | 21 (100%) | 0.407 | 0.3 | 56 (93%) |
| | Pericardial demarcation, n (%) | 36 (92%) | 21 (100%) | 0.347 | 0.5 | 57 (95%) |
| | Interatrial septal visualization, n (%) | 23 (59%) | 15 (71%) | 0.253 | 0.3 | 38 (63%) |

(*Continued*)

**Table 1.** (Continued)

| Characteristic | | Control, N = 39 | Quality indicator N = 21 | Cohen's D | p-value | Overall, N = 60 |
|---|---|---|---|---|---|---|
| **IVC view** | Total score, Mean (SD), (N) | 1.44 (0.72), (39) | 1.76 (0.44), (21) | 0.507 | 0.09 | 1.55 ± 0.65 (60) |
| | IVC visualization, n (%) | 25 (64%) | 18 (86%) | 0.478 | 0.08 | 43 (72%) |
| | RA, n (%) | 31 (79%) | 19 (90%) | 0.289 | 0.5 | 50 (83%) |

Abbreviations: LV- left ventricle; LA–left atrium; RV right ventricle; IVC–inferior vena cava

indicating tool and the non-AI group. Apart from using the quality indicator when practicing the apical 4- and 5-chamber views, both groups underwent identical hands-on training courses, including 4 hours of bedside teaching by experienced POCUS instructors. After the formal training hours, the students had access to the POCUS training lab which held ultrasound devices with and without the AI-based tool according to their study group. A healthy model was used during the course's hands-on, bedside teaching sessions, as well as during the exam.

## The AI quality indicator tool

The AI-based quality indicator tool provides real-time, three scale feedback on the quality of the apical 4-chamber image. This tool is part of the automatic EF tool. It presents the user with an LV endocardial border contour line that appears when the AI tool recognizes the apical 4- or 5-chamber views (Fig 1). The contour line can appear in green (best quality), yellow (medium quality), or red (unrecognized structures), according to the image quality determined by the AI algorithm. The algorithm analyzes real-time image quality, anatomical landmark identification, and EF result consistency (Fig 1). As mentioned, the intervention group used the tool during the course, additional training time, and exam. The control group did not use the tool at any point. All students were encouraged to practice outside the eight-hours course but were not mandated to do so. Students were required to document any extra practice hours for later analyses and comparisons between the groups.

## The six-minute exam

At the end of the course, we evaluated the students' POCUS handling skills with the previously established six-minute exam (see exam scoring criteria, S2 Appendix) [11]. This exam evaluates skills in obtaining key transthoracic cardiac views. The exam required students from both groups to acquire and store images of identical cardiac views.

Experienced POCUS instructors supervised the exam. Each student had 6 minutes to obtain clips with the POCUS device in a predetermined order of the following views: parasternal long axis; parasternal short axis including aortic valve (AV), mitral valve (MV), and mid-papillary level; apical 4-chamber; apical 5-chamber; apical 2-chamber; subcostal long axis and the inferior vena cava (IVC). All recorded clips were digitally stored for analysis.

After data collection was completed, the blinded review of the clips began. Clips were scrambled, and when reviewed, the quality indicator marker was removed to prevent the identification of the study group by the reviewer. A senior intensive care physician with over ten years of cardiac ultrasound experience performed a blinded rating of clip quality (grading anatomical landmarks acquisition and image quality). The exam score was based on a checklist of anatomical landmarks depicted in each echocardiographic view for a total of 31 points with one point per landmark (see scoring criteria in the exam, S2 Appendix). Models

passed a pretest screening for approval of the cardiac sonographic windows to reduce a bias of models' anatomical differences.

Additionally, we added another assessment criterion: the image quality score. This was a subjective criterion graded by the senior physician who blindly reviewed and graded all clips based on clinical experience and without restriction to specific landmarks. Overall scan quality scores were given as follows: 0—impossible to ascertain; 1- medium quality, readable image; and 2- good or excellent image.

## Statistical analysis

We compared the intervention and control groups' scores, using non-parametric Mann-Whitney U tests to determine significant differences. We analyzed specific views and quality assessments using t-tests and chi-square tests as appropriate. A Poisson regression analysis assessed the relative risk of achieving a higher than the median score, adjusting for relevant covariates. Furthermore, we calculated Cohen's D to measure the effect size, providing us with a standardized measure of the magnitude of the observed effects and aiding in interpreting the practical significance of our findings. All analyses were performed using R software version 4.0.2 (R Foundation for Statistical Computing, https://www.R-project.org/). We considered P values $< 0.05$ as statistically significant. Subgroup analyses were performed for gender, ethnicity, age, practice hours, and anatomy exams using the same tests and models.

## Results

A total of 100 students participated in the ultrasound training with 40 excluded from our study. Students were excluded if they did not sign the informed consent (n = 4), did not follow group allocation (n = 5), exceeded the 6-minute time limit (n = 15), scored below 15/31 points (n = 7), or a combination of these (n = 7). Details of exclusion according to group allocation are seen in Fig 2. We added the total score exclusion criterion since it was difficult to ascertain which views they attempted to achieve, rendering the data collected impossible to use. This was the case with 3/33 (9%) of the intervention group students and 11/63 (17.5%) of the control group students. There were no significant differences between the study groups in basic demographic parameters such as sex, age, ethnic background, anatomy exam score, and additional practice hours [Table 1].

## Total score

We found a significant difference between the groups in the total exam score. The intervention group had a median score of 26/31 (83.9%) as compared to 22/31 (71%) in the control group (P = 0.002). Cohen's D for the difference in total exam score between the groups was 0.890 [Table 2].

## Specific view scores

There were two specific views with significantly different student scores. The first was the parasternal long-axis view with a mean score of 3.48/4 in the intervention group and 2.77/4 in the control group (P = 0.01). Cohen's D for the difference in the long axis mean score between the groups was 0.536. In this view, the AI-based quality indicator does not function. Median scores were 4 and 3, respectively [Table 1]. The second was the apical 5-chamber view, where the AI-based quality indicator does function, with a mean score of 5.33/6 in the intervention group and 4.21/6 in the control group (P = 0.03) [Table 2]. Cohen's D for the difference in the apical 5-chamber mean score between the groups was 0.634. In both parasternal long and

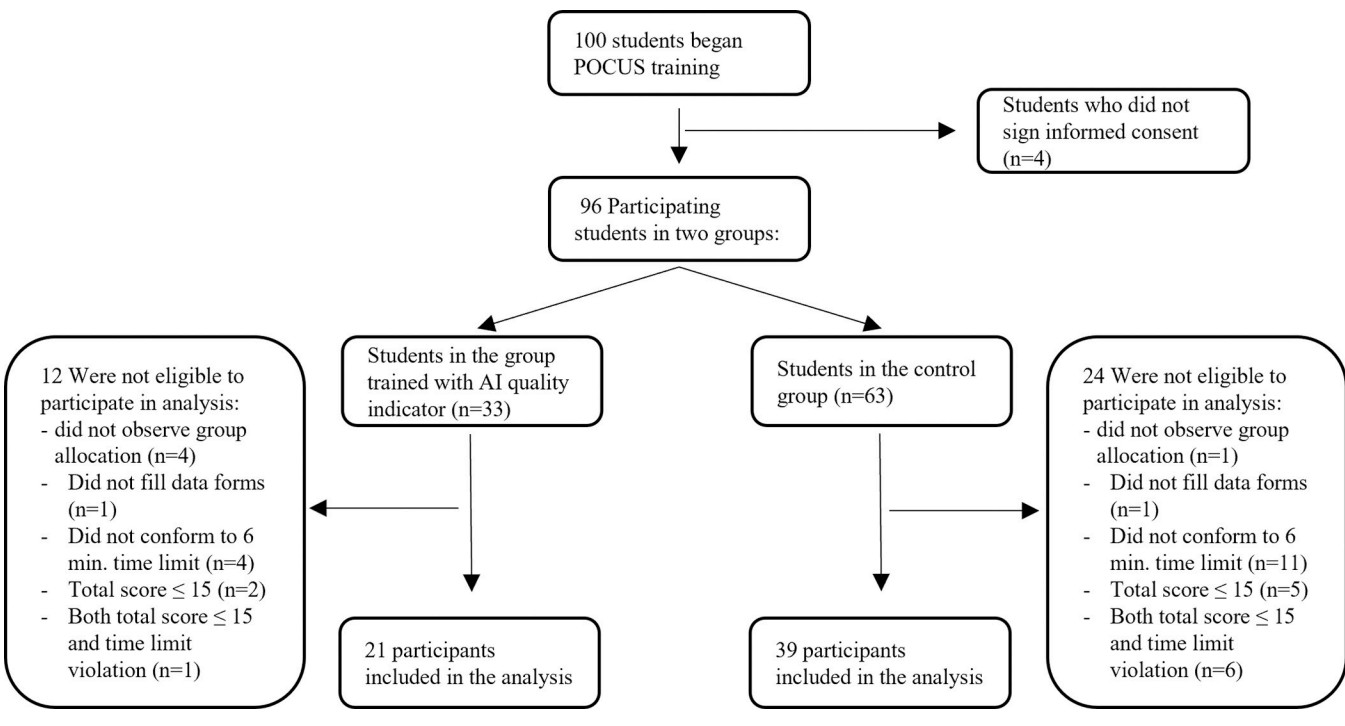

**Fig 2. A flowchart of the study and participant inclusion and exclusion.**

**Table 2. Comparison of success rates and exam scores between control group and quality indicator groups (apical 4/5-chamber included), a total of 60 participants, Ben Gurion University, Israel, 2022.**

| | Characteristic | Control, N = 39 | Quality indicator N = 21 | Cohen's D | p-value | Overall, N = 60 |
|---|---|---|---|---|---|---|
| | **Total exam score** (max. 31), Median (IQR) | 22.0 (19.5, 25.0) | 26.0 (24.0, 27.0) | 0.890 | **0.002** | 23.0 (21.0, 26.0) |
| **Apical 5-chamber view** | Total score, Mean (SD), (N) | 4.21 (2.05), (39) | 5.33 (0.97), (21) | 0.634 | **0.03** | 4.60 ± 1.82 (60) |
| | Open LV, n (%) | 30 (77%) | 20 (95%) | 0.491 | 0.08 | 50 (83%) |
| | RV visualization, n (%) | 27 (69%) | 18 (86%) | 0.376 | 0.2 | 45 (75%) |
| | Mitral Valve anatomy, n (%) | 32 (82%) | 20 (95%) | 0.383 | 0.2 | 52 (87%) |
| | Tricuspid anatomy, n (%) | 19 (49%) | 16 (76%) | 0.561 | **0.04** | 35 (58%) |
| | Open atrium, n (%) | 27 (69%) | 19 (90%) | 0.502 | 0.11 | 46 (77%) |
| | Aortic valve, n (%) | 29 (74%) | 19 (90%) | 0.398 | 0.2 | 48 (80%) |
| **Apical 4-chamber view** | Total score, Mean (SD), (N) | 4.44 (0.91), (39) | 4.43 (0.81), (21) | 0.008 | 0.8 | 4.43 ± 0.87 (60) |
| | Open LV, n (%) | 38 (97%) | 21 (100%) | 0.195 | >0.9 | 59 (98%) |
| | RV visualization, n (%) | 33 (85%) | 17 (81%) | 0.095 | 0.7 | 50 (83%) |
| | Mitral Valve anatomy, n (%) | 39 (100%) | 20 (95%) | 0.367 | 0.4 | 59 (98%) |
| | Tricuspid anatomy, n (%) | 30 (77%) | 16 (76%) | 0.017 | >0.9 | 46 (77%) |
| | Open atrium, n (%) | 33 (85%) | 18 (86%) | 0.030 | >0.9 | 51 (85%) |
| **Apical 4-chamber view** | Time to clip[a] (minutes) Mean (SD), (N) | 0.79 (0.56), (39) | 1.04 (1.35), (20) | 0.267 | 0.6 | 0.88 ± 0.90 (59) |

[a] Time between recording the last clip in sequence before apical 4-chamber, and the optimal clip for apical 4-chamber view

Abbreviations: LV—left ventricle; RV- right ventricle

apical 5-chamber views, all landmarks received higher success rates in the intervention group. The intervention group exhibited a trend of higher performance scores in most other cardiac windows but without statistical significance. In parasternal short views, the mean scores for the intervention and control group were as follows: aortic valve level view 3.05/4 and 2.54/4, respectively (P = 0.13); mitral valve level view 1.48/2 and 1.68/2, respectively (P = 0.4); and mid-papillary level view 1.33/2 for both groups (P>0.9) [Table 1]. Subcostal view mean scores were 2.71/3 and 2.41/3, respectively (P = 0.3); and IVC view mean scores were 1.76/2 and 1.44/2 (P = 0.09). Interestingly, we found no statistically significant differences between the groups in the apical 4-chamber view in which together with apical 5 chamber, the AI tool was designed to function. The mean scores were almost identical, 4.43/5 and 4.44/5 in the intervention and control groups, respectively (P = 0.8).

## Quality assessment scores

Blinded overall scan quality assessment scores generated significant differences in favor of the AI-based tool in various cardiac views. Quality indicator users achieved a higher score in the parasternal long axis (P = 0.002), apical 4-chamber (P = 0.003), apical 5-chamber (P = 0.003), subcostal (P = 0.04), and IVC (P = 0.02) views [Table 3]. Cohen's D for the difference in the quality assessment score between the groups was 0.815 for parasternal long axis, 0.840 for apical 4-chamber, 0.850 for apical 5-chamber, 0.594 in subcostal, and 0.631 for IVC view. We found insignificant differences in the parasternal short aortic valve level (P = 0.1), mid-papillary level (P>0.9), mitral valve level (P>0.9), and apical 2-chamber (P = 0.2) [Table 3].

## Time

The utilization of the automatic tool during the cardiac study did not prolong the scanning time for the clip it was applied to. There was no significant difference in the time required for apical 4-chamber acquisition (measured as the time elapsed between the last clip prior to the apical 4-chamber clip and the time of the optimal apical 4-chamber clip) (P = 0.6). The total test time of the study groups was equal.

**Table 3. Comparison of mean scores of subjective blinded quality assessment between control group and quality indicator users, a total of 60 participants, Ben Gurion University, Israel, 2022.**

| Echocardiographic view | Control N = 39 | Quality indicator N = 21 | P-value | Cohen's D | Overall, N = 60 |
|---|---|---|---|---|---|
| Parasternal long Mean (SD), (N) | 1.05 (0.60), (39) | 1.57 (0.68), (21) | **0.002** | 0.815 | 1.23 ± 0.67 (60) |
| Parasternal short, base Mean (SD), (N) | 0.92 (0.58), (39) | 1.19 (0.60), (21) | 0.10 | 0.449 | 1.02 ± 0.60 (60) |
| Parasternal short, mitral Mean (SD), (N) | 1.18 (0.64), (39) | 1.14 (0.85), (21) | >0.9 | 0.050 | 1.17 ± 0.72 (60) |
| Parasternal short, mid-papillary Mean (SD), (N) | 1.05 (0.79), (39) | 1.05 (0.92), (21) | >0.9 | 0.004 | 1.05 ± 0.83 (60) |
| Apical 4-chamber Mean (SD), (N) | 1.08 (0.35), (39) | 1.43 (0.51), (21) | **0.003** | 0.840 | 1.20 ± 0.44 (60) |
| Apical 5-chamber Mean (SD), (N) | 0.97 (0.54), (39) | 1.43 (0.51), (21) | **0.003** | 0.850 | 1.13 ± 0.57 (60) |
| Apical 2-chamber Mean (SD), (N) | 0.74 (0.75), (39) | 1.05 (0.80), (21) | 0.2 | 0.390 | 0.85 ± 0.78 (60) |
| Subcoastal Mean (SD), (N) | 1.18 (0.60), (39) | 1.52 (0.51), (21) | **0.04** | 0.594 | 1.30 ± 0.59 (60) |
| IVC, Mean (SD), (N) | 0.92 (0.74), (39) | 1.38 (0.67), (21) | **0.02** | 0.631 | 1.08 ± 0.74 (60) |

Abbreviations: IVC–inferior vena cava

## Subgroup analysis

We conducted subgroup analyses comparing exam scores and image quality in subgroups based on gender, ethnicity, age, and anatomy exam. We also tested the extracurricular training by students as a covariate for better performance and found it to be equally distributed between the two study groups (Table 1). Our analysis did not reveal any statistically significant differences between the two groups, leaving the automatic tool as the only significant factor that can explain score differences.

## Multivariate analysis

The Poisson regression analysis showed that quality indicator users had a relative risk of 2.3 (95% CI: 1.10, 4.93, P = 0.03) for receiving an overall score higher than the median score of 23 compared to the control group.

## Discussion

### 6-minute exam

The overall 6-minute exam score showed a significant advantage for the intervention group. This general improvement can be broken down into the direct effect of the quality indicator demarcation seen in apical 4- and 5-chamber views and the indirect effect when the quality indicator demarcation was absent in other views. Furthermore, the quality of cardiac views, the anatomical landmarks acquired, and the acquired image quality received significantly higher scores among the intervention compared to control students in most cardiac views, even in cardiac views where the quality indicator demarcation is not active [Table 3].

Nevertheless, although the quality indicator was designed for the apical 4-chamber view, the mean score in this view was not significantly higher in the intervention group. However, the apical 5-chamber, also a compatible view for the tool, received significantly higher scores among the intervention group. A more thorough investigation revealed that the control group lost points during the transition from apical 4-chamber view to apical 5-chamber (worsening LV imaging from 38 (97%) successful depictions to 30 (77%), RV from 33 (85%) to 27 (69%), MV from 39 (100%) to 32 (82%), tricuspid valve (TV) from 30 (77%) to 19 (49%), and atria from 33 (85%) to 27 (69%), respectively). In contrast to the control, the intervention group revealed identical or improved success rates of demonstrating cardiac structures when shifting from apical 4-chamber to apical 5-chamber (RV from 17 (81%) to 18 (86%), MV at 20 (95%) for both, TV at 16 (76%) for both, and atria from 18 (86%) to 19 (90%), respectively) [Table 2].

These results suggest that the quality indicator may assist in maintaining the cardiac landmarks shared by both views when the operator shifts from the apical 4- to 5-chamber view. This may be due to an advantageous probe positioning in the apical 4-chamber view while preparing to shift to apical 5-chamber. It is also possible that the quality indicator helped maintain the proper apical 5-chamber view landmarks despite the more complex positioning when presenting the aortic valve.

In most views other than the apical 4- and 5-chamber, the intervention group had higher mean scores than the control group, despite the tool not being active. These included the parasternal-long axis, parasternal short axis base, apical 2-chamber, subcostal, and IVC presenting views [Table 1]. The parasternal long axis is the only view with a statistically significant improvement. However, the overall mean test score was significantly higher for the intervention group, further supporting the generalized improvement in sonography skills.

Subgroup analysis results had no significant difference in exam scores between groups based on gender, ethnicity, age, self-practice hours outside of the course, and anatomy exam scores.

Our multivariate analysis, however, showed a relative risk of 2.3 (95% CI: 1.10, 4.93, P = 0.03) for higher than the median scores compared to the control group, adjusting for age and sex. This indicates that quality indicator use was associated with a significantly greater likelihood of obtaining a higher general score, attributing the difference between the group scores to the quality indicator.

## Quality assessment

We compared the quality of the cardiac images with the help of an independent, experienced clinician blinded to the study group. The blinding was done by removing the automatic EF LV demarcation line to prevent identification of quality indicator use. Unlike in the standardized 6-minute exam scoring, the clinician based it on his general impression, making it potentially the most difficult factor to predict in relation to the quality indicator. The intervention group achieved significant improvement in image quality compared to the control group. In 5 of 9 views, quality indicator users had statistically significant higher image quality grades. The advantageous views included apical 4- and 5-chamber, parasternal long, subcostal, and IVC views. Our explanation for the improved image quality, even in the views without the quality indicator use, is that it assists the users on several levels. First is gaining a better sense of the correct pressure applied on the probe for optimizing image quality. Second is the familiarity with the proper representation of the cardiac images, exposing them to better image standards that later helped them acquire more explicit images in the tool-free cardiac views. Additional factors may be that rather than identifying the correct general positioning and moving on, students using the quality indicator learn to take their time until optimizing the image. This could be done by optimizing patient positioning, adding ultrasound gel etc. This habit will more likely be acquired when receiving repeated feedback requiring quality improvement and landmark depiction. This can also be seen in the time students required for image acquisition, which although not statistically significant, was longer in the intervention group than the control group (1.04 and 0.79 minutes respectively, P = 0.6) [Table 2]. An additional factor which may have had an effect is the fact that the intervention group had a higher number of extracurricular training hours [Table 1]. However, this difference was statistically insignificant, and likely a less probable explanation.

Subgroup analysis did not show a significant difference in image quality scores between groups based on gender, ethnicity, age, practice hours, and anatomy exam scores, supporting the attribution of the improved score to the quality indicator.

These findings suggest a general effect of improved image acquisition that is not limited to the direct effect of the quality indicator demarcation. Previous studies on AI-assisted echocardiography often focused on software used for retrospective image analysis rather than real-time image acquisition and quality improvement [13–16]. In studies where image acquisition was tested, the focus was on comparing novice users to experienced clinicians [9, 10, 16]. Unlike previous studies, this study takes the assisting technology to the realm of medical education, comparing traditional learning methods with a new, real-time, AI-assisted tool. Our research suggests that real-time based feedback for cardiac ultrasound image quality improves image accuracy and quality among inexperienced users. As cardiac POCUS becomes an integral part of the standard physical examination, the demand for higher proficiency will enhance the need and development of such learning tools that enable novice operators to perform high-quality POCUS examinations.

Our research reinforces the established notion that interactive AI-based feedback tools can enhance POCUS performance, particularly in pulmonary and cardiovascular assessments [17,

18]. Our study demonstrates that incorporating AI live feedback tools during cardiac POCUS training can significantly enhance training efficiency and outcomes among medical students.

We believe the implications of our findings are twofold: first, in training for quality control, establishing professional standards for student training; and second, in the clinical field, as previous research shows that POCUS integrated automated AI tool integration can support and validate clinical decision-making, particularly among less experienced users [19]. Regarding the cardiac ultrasound applications explored in this study, we advocate for the integration of these tools into new POCUS systems and in student POCUS training programs. They prove invaluable in helping novice users optimize their imaging capabilities and further enhance the skills of experienced operators. Importantly, our findings indicate that the use of AI-based quality indicator tools did not significantly prolong the time required for image acquisition.

## Limitations

Our research has several limitations. The first and perhaps the most significant is that the 6-minute cardiac ultrasound competency exam took place immediately after a brief academic course. This is important as long-term differences (weeks or months after the course rather than days) between the groups are crucial in estimating effective medical education. Another significant limitation is the exclusion of over one-third of the participants from the study that could have caused a selection bias. It is important to stress that this exclusion was due to several reasons, as detailed in the results section. Criteria such as exceeding the time limit made the results incomparable to other users because time is an important factor in the exam, and the exclusion of students who scored ≤15 points was required because these students also had unclear images in their files, to such an extent that it was impossible to ascertain which views they were attempting to capture. Importantly, the excluded students were relatively equally divided between the two groups (Fig 2). Additionally, we tested the intervention group while using the quality indicator. This fact precludes our reporting on whether the tool improved performance in the intervention group even when not using the tool. However, for views other than the apical 4- and 5- chamber (where the tool is not active), results were better in the intervention group.

The study took place in a single university medical school, causing a possible selection bias. The sample may not represent the larger population since the medical school curriculum may differ between institutions. Thus, it may not accurately reflect the tool's advantage in other medical schools or healthcare settings. Further studies should be conducted on the intervention group without the tool in the future and should include larger sample sizes and additional medical schools.

## Conclusions

An AI-based quality indicator integrated into POCUS cardiac views improved the performance of cardiac ultrasound, as measured by the 6-minute exam, among recently trained students compared to students who did not use the tool. Improved scores were observed among the intervention group, even in cardiac views where the automatic tool was inactive. Such tools can assist in the learning process of cardiac ultrasound and should be integrated in new POCUS systems, in addition to expanding their repertoire to include more cardiac views. Further studies should be conducted to estimate their long-term effects on learning.

## Supporting information

**S1 Appendix. Research questionnaire.**
(PDF)

**S2 Appendix. 6-Minute exam scoring.**
(PDF)

**S1 Data set.**
(XLSX)

# Acknowledgments

This research is part of the qualification requirements for M.D. approval at the Joyce & Irving Goldman Medical School at the Faculty of Health Sciences, Ben-Gurion University of the Negev, Israel.

# Author Contributions

**Conceptualization:** Noam Aronovitz, Itai Hazan, Roni Jedwab, Lior Fuchs.

**Data curation:** Noam Aronovitz, Roni Jedwab, Lior Fuchs.

**Formal analysis:** Itai Hazan, Lior Fuchs.

**Investigation:** Noam Aronovitz, Itai Hazan, Roni Jedwab, Lior Fuchs.

**Methodology:** Noam Aronovitz, Itai Hazan, Lior Fuchs.

**Project administration:** Noam Aronovitz, Itamar Ben Shitrit, Oren Wacht.

**Resources:** Oren Wacht, Lior Fuchs.

**Software:** Itai Hazan.

**Supervision:** Lior Fuchs.

**Writing – original draft:** Noam Aronovitz, Itai Hazan, Anna Quinn, Lior Fuchs.

**Writing – review & editing:** Noam Aronovitz, Itai Hazan, Itamar Ben Shitrit, Anna Quinn, Lior Fuchs.

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
