## [Decision Letter · Decision Letter 0]

29 Aug 2023

PONE-D-23-23122The effect of real-time EF automatic tool on cardiac ultrasound performance among medical studentsPLOS ONE

Dear Dr. ARONOVITZ,

Thank you for submitting your manuscript to PLOS ONE. After careful consideration, we feel that it has merit but does not fully meet PLOS ONE’s publication criteria as it currently stands. Therefore, we invite you to submit a revised version of the manuscript that addresses the points raised during the review process.

We look forward to receiving your revised manuscript.

Kind regards,

Vikramaditya Samala Venkata

Academic Editor

PLOS ONE

Journal Requirements:

2. Please note that PLOS ONE has specific guidelines on code sharing for submissions in which author-generated code underpins the findings in the manuscript. In these cases, all author-generated code must be made available without restrictions upon publication of the work. 

Please review our guidelines at https://journals.plos.org/plosone/s/materials-and-software-sharing#loc-sharing-code and ensure that your code is shared in a way that follows best practice and facilitates reproducibility and reuse.

"I have read the journal's policy and the authors of this manuscript have the following competing interests: 

GE healthcare© provided the POCUS devices used for this study. 

Lior Fuchs declares that he

works as a consultant for GE healthcare. The company had no access to the idea, to the study's primary objective, nor to its design, data analysis or writing. 

The rest of the authors declare that

no competing interests exist."

We note that one or more of the authors are employed by a commercial company: GE healthcare

(2) Please also provide an updated Competing Interests Statement declaring this commercial affiliation along with any other relevant declarations relating to employment, consultancy, patents, products in development, or marketed products, etc.  

Within your Competing Interests Statement, please confirm that this commercial affiliation does not alter your adherence to all PLOS ONE policies on sharing data and materials by including the following statement: "This does not alter our adherence to  PLOS ONE policies on sharing data and materials.” (as detailed online in our guide for authors http://journals.plos.org/plosone/s/competing-interests). 

If this adherence statement is not accurate and  there are restrictions on sharing of data and/or materials, please state these. Please note that we cannot proceed with consideration of your article until this information has been declared.

5. We note that Figure 1 in your submission contain copyrighted images. All PLOS content is published under the Creative Commons Attribution License (CC BY 4.0), which means that the manuscript, images, and Supporting Information files will be freely available online, and any third party is permitted to access, download, copy, distribute, and use these materials in any way, even commercially, with proper attribution. For more information, see our copyright guidelines: http://journals.plos.org/plosone/s/licenses-and-copyright.

(1) You may seek permission from the original copyright holder of Figure 1 to publish the content specifically under the CC BY 4.0 license. 

(2) If you are unable to obtain permission from the original copyright holder to publish these figures under the CC BY 4.0 license or if the copyright holder’s requirements are incompatible with the CC BY 4.0 license, please either i) remove the figure or ii) supply a replacement figure that complies with the CC BY 4.0 license. Please check copyright information on all replacement figures and update the figure caption with source information. 

If applicable, please specify in the figure caption text when a figure is similar but not identical to the original image and is therefore for illustrative purposes only.

was small in both groups)

**Additional Editor Comments:**

1)
Can authors explain if this AI indicator is available/or will be available with other brand of ultrasound machines. Basically can these results be reproduced at other institutions with other brand machines. What is the cost component with this AI indicator.2)
As reviewers note. Even for non apical views without AI tool. Intervention group had better scores. Authors proposed a reason stating, indicator can help with probe pressure etc  Can authors propose an alternative reason for this? The mean and median number of extracurricular hours were more in the intervention group in table 1( I understand it was not statistically significant but at the same time, N was small in both groups)3)
Whats the main take home point from this study/ or aim. Should we incorporate this specific AI tool in medical school education or should we start using ultrasound companies which have this new AI tool etc ( as reviewer mentions below, we need to go beyond presenting the results, explain what these results mean for the field of medicine in more detail)4)
Please see reviewer comments below and address them

Reviewers' comments:

Reviewer's Responses to Questions

**Comments to the Author**

1. Is the manuscript technically sound, and do the data support the conclusions?

Reviewer #1: Yes

Reviewer #2: Yes

Reviewer #3: Yes

Reviewer #4: Yes

Reviewer #5: Yes

2. Has the statistical analysis been performed appropriately and rigorously? 

Reviewer #1: Yes

Reviewer #2: Yes

Reviewer #3: Yes

Reviewer #4: Yes

Reviewer #5: Yes

3. Have the authors made all data underlying the findings in their manuscript fully available?

Reviewer #1: Yes

Reviewer #2: Yes

Reviewer #3: Yes

Reviewer #4: Yes

Reviewer #5: Yes

4. Is the manuscript presented in an intelligible fashion and written in standard English?

Reviewer #1: Yes

Reviewer #2: Yes

Reviewer #3: Yes

Reviewer #4: Yes

Reviewer #5: Yes

5. Review Comments to the Author

Reviewer #1: While the introduction highlights the importance of POCUS and the challenges in integrating it into medical education, it lacks a clear statement of the research problem or hypothesis. What specific aspect are you trying to address or investigate with your study?

The recruitment process for participants could be described in more detail. How were students selected, and what criteria were used for exclusion?

It would be helpful to provide effect sizes or confidence intervals for the statistically significant differences you've identified.

The discussion section lacks depth in analyzing the results and their implications. It's important to go beyond just presenting the results and explain what they mean for the field of medical education, POCUS training, and the use of AI tools.

Reviewer #2: Any reason why only male students were enrolled in the study?

Only one experienced provider has reviewed the result. Study results reviewed by multiple provider might have provided more reliability.

Did the subjects in this study have any cardiac pathologies?

Did case and control groups have same subjects? If subjects differ, which group had more patients with cardiac pathologies?

Reviewer #3: Overall Assessment

The study is well-conceived and addresses an important issue in medical education, specifically focusing on the impact of an AI-based quality indicator tool in teaching cardiac point-of-care ultrasound (POCUS) to 4th-year medical students. The research methodology is sound, and the results are statistically significant, suggesting that the AI-based tool does improve both the performance and image quality in cardiac ultrasound.

Strengths

The topic is relevant to medical education and has implications for better diagnostic performances, especially in point-of-care settings.

The study design is robust, including a control group for comparison.

Detailed statistical analysis strengthens the reliability of the findings, and the authors controlled for multiple covariates.

Areas for Improvement

Long-term Retention: The study's most significant limitation, as pointed out by the authors, is the immediate testing after the academic course. An evaluation of long-term retention of skills would provide a more comprehensive understanding of the tool's effectiveness.

Selection Bias: Another limitation is that over a third of the participants were excluded from the study, potentially causing a selection bias. This needs to be addressed more in the discussion.

Single Institution Study: The study being conducted at a single medical school may not generalize to other settings. Multi-center studies are encouraged.

Tool Availability: The intervention group was tested using the quality indicator, so it is not clear if the skills were internalized enough to maintain performance without the tool. Future studies could look into this.

The study appears to adhere to ethical guidelines, including informed consent and approval by an ethics committee. However, the authors should clarify whether all students were offered the chance to benefit from the AI tool post-study, given its proven efficacy.

Publication Ethics

No concerns about dual publication or publication ethics are apparent from the manuscript.

Reviewer #4: 1. The research involves assessing skills learned within hours which makes it difficult to avoid operator bias, as few students might be better than others or some do better by chance. However the use of the 6 mins test is a better idea to overcome the issue.

2. It is also difficult to see how this will help in long term practice as the operator of POCUS become more experienced the quality of obtaining images is improves. So, hard to justify the absolute necessity of the AI tools but definitely it will help learning at beginners level.

3. Over all good study but practical use ll be limited.

Reviewer #5: I congratulate the investigators. Overall, it is a good study and well-written.

There are some minor revision recommendations.

1. Please provide the supplementary document of the quality assessment.

2. Line number 55-56, please rewrite it; it needs to be clarified.

3. It's better to report the OR rather than the RR.

6. PLOS authors have the option to publish the peer review history of their article (what does this mean?). If published, this will include your full peer review and any attached files.

Reviewer #1: No

Reviewer #2: No

Reviewer #3: **Yes: **Naveen Prasath Baskaran

Reviewer #4: **Yes: **Nihar Jena

Reviewer #5: No

---

## [Author Response · Author response to Decision Letter 0]

11 Dec 2023

Dear editor and reviewing team, 

Thank you for your feedback. In the following response to the reviewers, we will address the points raised by the academic editor and reviewers. We would like to thank you for an in-depth assessment and for your important remarks. Careful consideration was given to address these comments. In this letter you will find a point-by-point response to your comments and suggestions. Where it was relevant, we have added line numbered references to changes made in the manuscript (the numbering applies to the untracked version). 

Once again, we would like to thank you all for your attention and the energy invested in this review. We have taken extensive measures to address the concerns you have shared with us and are confident our work now offers a coherent and valuable contribution to the scientific community. 

With thanks, 

Noam Aronovitz, Itai Hazan, Roni Jedwab, Itamar Ben Shitrit, Anna Quinn, Oren Wacht, and Lior Fuchs 

Responses to the editor: 

Additional requirements 

Comment: Please ensure that your manuscript meets PLOS ONE's style requirements, including those for file naming 

Response: We have ensured that our manuscript meets PLOS ONE’s style requirements, including file naming. 

Comment: Please note that PLOS ONE has specific guidelines on code sharing for submissions in which author-generated code underpins the findings in the manuscript. In these cases, all author-generated code must be made available without restrictions upon publication of the work. 

Response: There has been no author generated code in the process of our study, except for the code used for statistical analysis when using the “R” program. We have added the “syntax.R” file as “other” assuming this is the code you were referring to in your comment. 

Comment: (1) Please provide an amended Funding Statement declaring this commercial affiliation, as well as a statement regarding the Role of Funders in your study. If the funding organization did not play a role in the study design, data collection and analysis, decision to publish, or preparation of the manuscript and only provided financial support in the form of authors' salaries and/or research materials, please review your statements relating to the author contributions, and ensure you have specifically and accurately indicated the role(s) that these authors had in your study. You can update author roles in the Author Contributions section of the online submission form. 

Please also include the following statement within your amended Funding Statement: “The funder provided support in the form of salaries for authors [insert relevant initials], but did not have any additional role in the study design, data collection and analysis, decision to publish, or preparation of the manuscript. The specific roles of these authors are articulated in the ‘author contributions’ section.” 

If your commercial affiliation did play a role in your study, please state and explain this role within your updated Funding Statement.  

(2) Please also provide an updated Competing Interests Statement declaring this commercial affiliation along with any other relevant declarations relating to employment, consultancy, patents, products in development, or marketed products, etc.  Within your Competing Interests Statement, please confirm that this commercial affiliation does not alter your adherence to all PLOS ONE policies on sharing data and materials by including the following statement: "This does not alter our adherence to PLOS ONE policies on sharing data and materials.” (as detailed online in our guide for authors http://journals.plos.org/plosone/s/

competing-interests). If this adherence statement is not accurate and  there are restrictions on sharing of data and/or materials, please state these. Please note that we cannot proceed with consideration of your article until this information has been declared. 

Response: An updated funding statement and competing interests statement were added to our cover letter. 

Comment: In your Data Availability statement, you have not specified where the minimal data set underlying the results described in your manuscript can be found. PLOS defines a study's minimal data set as the underlying data used to reach the conclusions drawn in the manuscript and any additional data required to replicate the reported study findings in their entirety. All PLOS journals require that the minimal data set be made fully available. For more information about our data policy, please see http://journals.plos.org/plosone/s/data-availability. 

Upon re-submitting your revised manuscript, please upload your study’s minimal underlying data set as either Supporting Information files or to a stable, public repository and include the relevant URLs, DOIs, or accession numbers within your revised cover letter. For a list of acceptable repositories, please see http://journals.plos.org/plosone/s/data-availability#loc-recommended-repositories . Any potentially identifying patient information must be fully anonymized. 

Response: We have uploaded our study’s minimal data set as a Supporting Information file. 

Comment: We note that Figure 1 in your submission contains copyrighted images. All PLOS content is published under the Creative Commons Attribution License (CC BY 4.0), which means that the manuscript, images, and Supporting Information files will be freely available online, and any third party is permitted to access, download, copy, distribute, and use these materials in any way, even commercially, with proper attribution. For more information, see our copyright guidelines: http://journals.plos.org/plosone/s/licenses-and-copyright. 

Response: We have received written permission from the copyright holder (GE healthcare©) and are awaiting their legal team to send in the signed form. This may take a few more days (they committed to send us the signed form by Sunday 10.15.2023) and we would appreciate your understanding on this matter. Temporary written approval by email was uploaded as “other” with the description “Email approval from GE for image use”. 

Comment: Please review your reference list to ensure that it is complete and correct. If you have cited papers that have been retracted, please include the rationale for doing so in the manuscript text, or remove these references and replace them with relevant current references. Any changes to the reference list should be mentioned in the rebuttal letter that accompanies your revised manuscript. If you need to cite a retracted article, indicate the article’s retracted status in the References list and also include a citation and full reference for the retraction notice. 

Response: We have reviewed our reference list and ensured it is complete and in the correct format. Some changes were made to correct format and an additional reference was added (reference 19) while writing the revision. We have checked for retractions and none of the papers cited in our manuscript have been retracted. 

Additional editor comments: 

Comment: Can authors explain if this AI indicator is available/or will be available with other brand of ultrasound machines. Basically can these results be reproduced at other institutions with other brand machines. What is the cost component with this AI indicator 

Response: The question regarding availability of the AI indicator is an important one. The integration of AI technology can be seen in most of the major ultrasound manufacturers, and the automated ejection fraction measurement is part of that trend. The function can be inserted as an upgrade rather than a new system altogether. For the Venue Go system (the system we used in our study) the cost of the basic system is 28,000 USD. Adding the AI tool and auto EF function would cost an additional 3,500 USD. Another example can be found in Kosmos Trio by EchoNous© which has a very similar interface to the one we used in our study. The cost of their system is 10,000 USD without the AI tool, and an additional 2,000 USD would provide users with the automatic EF function which also comes with an AI quality indicator similar to the one we used in our study. These are expensive systems at base, but the addition of the AI component is relatively less expensive because it is usually a matter of software upgrading. Hopefully, as the use of quality indicators becomes more standard, we will see more companies integrating quality indicators in addition to automated measurements. We believe this will add a great deal of value in those places where manufacturers choose to do so. 

Comment: As reviewers note. Even for non apical views without AI tool. Intervention group had better scores. Authors proposed a reason stating, indicator can help with probe pressure etc Can authors propose an alternative reason for this? The mean and median number of extracurricular hours were more in the intervention group in table 1( I understand it was not statistically significant but at the same time, N was small in both groups) 

Response: We have expanded the discussion about alternative reasons for the improved scores among AI tool users. This is addressed in the manuscript in lines 310-323. Indeed, there is a difference in extracurricular training time; however, the statistical insignificance of these differences is not only due to a small N, but also due to the small absolute difference in training time which we found likely to be inconsequential. Despite this, we agree that it is noteworthy, and we have mentioned these differences in the aforementioned correction. 

Comment: What’s the main take home point from this study/ or aim. Should we incorporate this specific AI tool in medical school education or should we start using ultrasound companies which have this new AI tool etc (as reviewer mentions below, we need to go beyond presenting the results, explain what these results mean for the field of medicine in more detail) 

Response: We have expanded the discussion regarding the implications of our study results, and we have recommended the integration of POCUS AI tools in medical education. Corrections can be seen in lines 346-358. 

Reviewer comments 

Reviewer #1 

Comment: While the introduction highlights the importance of POCUS and the challenges in integrating it into medical education, it lacks a clear statement of the research problem or hypothesis. What specific aspect are you trying to address or investigate with your study? 

Response: We have clarified the research hypothesis and the goals of our study. Corrections can be found in lines 92-99. 

Comment: The recruitment process for participants could be described in more detail. How were students selected, and what criteria were used for exclusion? 

Response: We expanded the section describing the student recruitment process, describing student selection and exclusion criteria. Changes can be found in lines 105 -114. 

Comment: It would be helpful to provide effect sizes or confidence intervals for the statistically significant differences you've identified. 

Response: As recommended, we have calculated Cohen's D to measure the effect size, providing us with a standardized measure of the magnitude of the observed effects and aiding in interpreting the practical significance of our findings. Alterations were made accordingly throughout the manuscript, including a revised “Statistical analysis” paragraph, additions to tables 1, 2, and 3 and additional changes in the “Results” section. 

Comment: The discussion section lacks depth in analyzing the results and their implications. It's important to go beyond just presenting the results and explain what they mean for the field of medical education, POCUS training, and the use of AI tools. 

Response: Thank you for this important remark. We have expanded the discussion as recommended by both you and the editor (additional editor comment #3). Corrections can be seen in lines 346-358. 

Reviewer #2 

Comment:  Any reason why only male students were enrolled in the study? 

Response: As presented in Table1, there was an equal male:female ratio between the two groups, both male and female students participated in the study. 

Comment: Only one experienced provider has reviewed the result. Study results reviewed by multiple providers might have provided more reliability. 

Response: We agree that it would have been preferable to have more than one provider review the results, however our resources sufficed for only one. We believe that the fact that the results were presented blindly provided the necessary reliability. 

3,4. Comment: Did the subjects in this study have any cardiac pathologies? Did case and control groups have same subjects? If subjects differ, which group had more patients with cardiac pathologies? 

Response: None of the subjects in the study suffered from cardiac pathologies. The goal in this study was acquiring correct cardiac views and not testing diagnostic skills, although there is value in also introducing that function in further studies. The subjects in the study were randomly assigned to the groups: some were only in certain groups and some moved between groups as required by logistical constraints. As stated in the manuscript, all subjects passed a pretest screening for approving the cardiac sonographic windows to reduce potential bias due to anatomical differences. 

Reviewer #3 

We thank you for your positive overall assessment. Regarding specific remarks relating to areas for improvement: 

Comment: Long-term Retention: The study's most significant limitation, as pointed out by the authors, is the immediate testing after the academic course. An evaluation of long-term retention of skills would provide a more comprehensive understanding of the tool's effectiveness. 

Response: As pointed out in our study, we are aware of the study’s limitations in testing for long term retention. This question is beyond the scope of our current study, and its exploration was limited by course length and the availability of the relevant POCUS systems. 

 Comment: Selection Bias: Another limitation is that over a third of the participants were excluded from the study, potentially causing a selection bias. This needs to be addressed more in the discussion. 

Response: We are aware of the potential selection bias due to the exclusion criteria and excluded participants. We have elaborated our explanation addressing this issue in the manuscript. Changes can be found in lines 363-371. 

Comment: Single Institution Study: The study being conducted at a single medical school may not generalize to other settings. Multi-center studies are encouraged. 

Response: Regarding the possible bias due to our study being limited to a single institution, as pointed out in our study, we are aware of this limitation. However, a multi-center study exceeds the scope of our current work. 

Comment: Tool Availability: The intervention group was tested using the quality indicator, so it is not clear if the skills were internalized enough to maintain performance without the tool. Future studies could look into this. 

Response: We appreciate you stressing this point and strongly agree. Indeed, the question regarding long term internalization of acquired skills should be tested. We intend to address this question in further studies, as suggested. 

Comment: The study appears to adhere to ethical guidelines, including informed consent and approval by an ethics committee. However, the authors should clarify whether all students were offered the chance to benefit from the AI tool post-study, given its proven efficacy. 

Response: The POCUS systems that we used during the study were borrowed from the manufacturer for the purpose of the study and were thereafter returned to the manufacturer. The purpose of the study was to establish the advantage of using the AI tool and we intend to implement our findings by using this tool in the courses to come. However, the students will have an opportunity to use this tool in clinical rounds in the future, as these systems are being integrated into hospital wards in their training hospitals. 

Reviewer #4 

Comment: The research involves assessing skills learned within hours which makes it difficult to avoid operator bias, as few students might be better than others or some do better by chance. However the use of the 6 mins test is a better idea to overcome the issue. 

Response: We appreciate your relating to the difficulty with operator bias. Indeed, in the 6-minute exam we try to make as clear an assessment as possible, and together with the control group we believe the statistically significant results in our analysis provide compelling support for our hypothesis. 

Comment: It is also difficult to see how this will help in long term practice as the operator of POCUS become more experienced the quality of obtaining images is improves. So, hard to justify the absolute necessity of the AI tools but definitely it will help learning at beginners level. 

Response: We agree with the claim that our findings are more significant during training and among inexperienced users. As we have written in the introduction, the challenge we address is the time required for medical professionals to gain the important experience needed to properly operate the POCUS systems. Our study suggests that the AI tool can help reduce the time required to reach the proficiency level needed for clinical work, by allowing more efficient training, and can support users during the training time by providing them with a live feedback tool. 

Reviewer #5 

Comment: Please provide the supplementary document of the quality assessment. 

Response: A supplementary document with information regarding quality assessment scores has been submitted along with our “minimal data set”. Unlike the 6-minute exam there are no conventional landmarks provided, rather it represents a general impression as assessed by an intensive care specialist with extensive POCUS training and experience. 

Comment: Line number 55-56, please rewrite it; it needs to be clarified. 

Response: We have rephrased the lines you have related to. A corrected and clearer version can be found in lines 60-61. 

Comment: It's better to report the OR rather than the RR. 

Response: Thank you for this remark. After discussing this issue with our statistical consultants, we strongly believe the analysis in a study of this kind should report the RR. We have calculated the OR and believe it to be an overestimation (OR= 7.01, P= 0.003), as supported in previous studies related to the topic1,2. These studies recommend presenting the RR for RCTs and cohort studies, as the RR is less likely to overestimate the effect size. 

Additional changes made: 

Correction of author affiliations. 

Minimal changes in writing style and grammar throughout the manuscript, making the text more coherent. 

Added source of Fig 1 in reference list. 

Knol MJ, Duijnhoven RG, Grobbee DE, Moons KGM, Groenwold RHH. Potential Misinterpretation of Treatment Effects Due to Use of Odds Ratios and Logistic Regression in Randomized Controlled Trials. PLOS ONE. 2011;6(6):e21248. doi:10.1371/journal.pone.0021248. 

Knol MJ, Le Cessie S, Algra A, Vandenbroucke JP, Groenwold RH. Overestimation of risk ratios by odds ratios in trials and cohort studies: alternatives to logistic regression. CMAJ. 2012;184(8):895-9. doi:10.1503/cmaj.101715. Epub 2011 Dec 12. PMID: 22158397; PMCID: PMC3348192.

---

## [Decision Letter · Decision Letter 1]

12 Feb 2024

The effect of real-time EF automatic tool on cardiac ultrasound performance among medical students

PONE-D-23-23122R1

Dear Dr. ARONOVITZ,

We’re pleased to inform you that your manuscript has been judged scientifically suitable for publication and will be formally accepted for publication once it meets all outstanding technical requirements.

Kind regards,

Antoine Fakhry AbdelMassih

Academic Editor

PLOS ONE

Additional Editor Comments (optional):

Reviewers' comments:

Reviewer's Responses to Questions

**Comments to the Author**

1. If the authors have adequately addressed your comments raised in a previous round of review and you feel that this manuscript is now acceptable for publication, you may indicate that here to bypass the “Comments to the Author” section, enter your conflict of interest statement in the “Confidential to Editor” section, and submit your "Accept" recommendation.

Reviewer #1: All comments have been addressed

Reviewer #3: All comments have been addressed

2. Is the manuscript technically sound, and do the data support the conclusions?

Reviewer #1: Partly

Reviewer #3: Yes

3. Has the statistical analysis been performed appropriately and rigorously? 

Reviewer #1: Yes

Reviewer #3: Yes

4. Have the authors made all data underlying the findings in their manuscript fully available?

Reviewer #1: Yes

Reviewer #3: Yes

5. Is the manuscript presented in an intelligible fashion and written in standard English?

Reviewer #1: Yes

Reviewer #3: Yes

6. Review Comments to the Author

Reviewer #1: Excellent work on your article! It's clear you've thoroughly researched the topic and presented your findings effectively.

Reviewer #3: study's design and implementation are good, providing valuable insights into the effectiveness of real-time feedback mechanisms on learning outcomes. Additionally, addressing potential biases more thoroughly, ensuring data accessibility, and discussing ethical considerations related to AI use in education would further strengthen the manuscript. Overall, the study is a meaningful step towards integrating advanced technologies into medical training, with suggestions provided aiming to refine and enhance its contribution to the field.

7. PLOS authors have the option to publish the peer review history of their article (what does this mean?). If published, this will include your full peer review and any attached files.

Reviewer #1: No

Reviewer #3: No

---

## [Editor Report · Acceptance letter]

20 Mar 2024

PONE-D-23-23122R1 

PLOS ONE

Dear Dr. Aronovitz, 

I'm pleased to inform you that your manuscript has been deemed suitable for publication in PLOS ONE. Congratulations! Your manuscript is now being handed over to our production team.

Kind regards, 

on behalf of

Prof Antoine Fakhry AbdelMassih 

Academic Editor

PLOS ONE